# Genome-Wide Association Study of Metamizole-Induced Agranulocytosis in European Populations

**DOI:** 10.3390/genes11111275

**Published:** 2020-10-29

**Authors:** Anca Liliana Cismaru, Deborah Rudin, Luisa Ibañez, Evangelia Liakoni, Nicolas Bonadies, Reinhold Kreutz, Alfonso Carvajal, Maria Isabel Lucena, Javier Martin, Esther Sancho Ponce, Mariam Molokhia, Niclas Eriksson, Stephan Krähenbühl, Carlo R. Largiadèr, Manuel Haschke, Pär Hallberg, Mia Wadelius, Ursula Amstutz

**Affiliations:** 1Department of Clinical Chemistry, Inselspital Bern University Hospital, University of Bern, 3010 Bern, Switzerland; ancacismaru@aol.com (A.L.C.); carlo.largiader@insel.ch (C.R.L.); 2Graduate School for Cellular and Biomedical Sciences, University of Bern, 3012 Bern, Switzerland; 3Department of Clinical Pharmacology & Toxicology, University Hospital Basel, University of Basel, 4031 Basel, Switzerland; deborah.rudin@meduniwien.ac.at (D.R.); stephan.kraehenbuehl@unibas.ch (S.K.); 4Department of Biomedicine, University of Basel, 4051 Basel, Switzerland; 5Clinical Pharmacology Service, Hospital Universitari Vall d’Hebron, Department of Pharmacology, Therapeutics and Toxicology, Autonomous University of Barcelona, Fundació Institut Català de Farmacología, 08035 Barcelona, Spain; li@icf.uab.cat; 6Department of Clinical Pharmacology & Toxicology, Inselspital Bern University Hospital, University of Bern, 3010 Bern, Switzerland; evangelia.liakoni@insel.ch (E.L.); manuel.haschke@insel.ch (M.H.); 7Institute of Pharmacology, University of Bern, 3012 Bern, Switzerland; 8Department of Hematology and Central Hematology Laboratory, Inselspital Bern University Hospital, University of Bern, 3010 Bern, Switzerland; nicolas.bonadies@insel.ch; 9Charité–Universitätsmedizin Berlin, Corporate Member of Freie Universität Berlin, Humboldt-Universität zu Berlin, and Berlin Institute of Health, Institut für Klinische Pharmakologie und Toxikologie, 10117 Berlin, Germany; reinhold.kreutz@charite.de; 10Centro de Estudios sobre la Seguridad de los Medicamentos, Universidad de Valladolid, 47005 Valladolid, Spain; carvajal@ife.uva.es; 11Servicio Farmacologia Clinica, Instituto de Investigación Biomedica de Málaga, Hospital Universitario Virgen de la Victoria, Universidad de Málaga, 29010 Málaga, Spain; lucena@uma.es; 12Instituto de Parasitología y Biomedicina Lopez-Neyra, Consejo Superior de Investigaciones Cientiíficas, 18016 Granada, Spain; javiermartin@ipb.csic.es; 13Servei d’Hematologia i Banc de Sang, Hospital General de Catalunya, 08190 Sant Cugat del Vallès, Spain; esther.sancho@quironsalud.es; 14Department of Population Health Sciences, King’s College London, London WC2R 2LS, UK; mariam.molokhia@kcl.ac.uk; 15Uppsala Clinical Research Center and Department of Medical Sciences, Uppsala University, 751 85 Uppsala, Sweden; niclas.eriksson@ucr.uu.se; 16Department of Medical Sciences, Clinical Pharmacology and Science for Life Laboratory, Uppsala University, 751 85 Uppsala, Sweden; par.hallberg@medsci.uu.se (P.H.); mia.wadelius@medsci.uu.se (M.W.)

**Keywords:** drug-induced agranulocytosis, metamizole, dipyrone, genome-wide association study, pharmacogenetics

## Abstract

Agranulocytosis is a rare yet severe idiosyncratic adverse drug reaction to metamizole, an analgesic widely used in countries such as Switzerland and Germany. Notably, an underlying mechanism has not yet been fully elucidated and no predictive factors are known to identify at-risk patients. With the aim to identify genetic susceptibility variants to metamizole-induced agranulocytosis (MIA) and neutropenia (MIN), we conducted a retrospective multi-center collaboration including cases and controls from three European populations. Association analyses were performed using genome-wide genotyping data from a Swiss cohort (45 cases, 191 controls) followed by replication in two independent European cohorts (41 cases, 273 controls) and a joint discovery meta-analysis. No genome-wide significant associations (*p* < 1 × 10^−7^) were observed in the Swiss cohort or in the joint meta-analysis, and no candidate genes suggesting an immune-mediated mechanism were identified. In the joint meta-analysis of MIA cases across all cohorts, two candidate loci on chromosome 9 were identified, rs55898176 (OR = 4.01, 95%CI: 2.41–6.68, *p* = 1.01 × 10^−7^) and rs4427239 (OR = 5.47, 95%CI: 2.81–10.65, *p* = 5.75 × 10^−7^), of which the latter is located in the *SVEP1* gene previously implicated in hematopoiesis. This first genome-wide association study for MIA identified suggestive associations with biological plausibility that may be used as a stepping-stone for post-GWAS analyses to gain further insight into the mechanism underlying MIA.

## 1. Introduction

Metamizole, or dipyrone, is an analgesic and antipyretic drug used to manage different types of pain as well as fever and often serves as an alternative to therapy with conventional non-steroidal anti-inflammatory drugs (NSAIDs). Despite a well-known favorable gastrointestinal safety profile [1,2,3], spontaneous adverse drug reaction (ADR) reports with risk estimates varying between different countries [4,5,6] provide accounts of metamizole-induced blood dyscrasias. Specifically, these include metamizole-induced neutropenia (MIN) and a more severe form, agranulocytosis (MIA) characterized by a decrease in circulating neutrophil granulocytes below 1.5 × 10^9^ cells/L, and 0.5 × 10^9^ cells/L, respectively [7,8]. Consequently, the use of metamizole is currently subjected to contrasting regulations put in place by different government authorities, ranging from market withdrawal to non-prescription use [4,5,9,10,11,12]. To date, the pathogenesis and biologic markers of risk for metamizole-induced agranulocytosis (MIA) remain poorly understood and there are no effective strategies for prediction or prevention.

Genome-wide association studies (GWAS) have served as a cost-effective approach to identify associations of single nucleotide polymorphisms (SNPs) with a variety of human phenotypes ranging from complex diseases to drug-related outcomes. In particular, previous GWAS have identified associations of genetic variants with rather large effect sizes for the susceptibility to various rare and serious adverse drug reactions (ADRs) such as flucloxacillin-induced liver injury, statin-induced myopathy and carbamazepine-induced skin injury [13,14]. In more recent years, studies of idiosyncratic drug-induced agranulocytosis identified genes associated with increased risk either in the human leukocyte antigen (HLA) region or in other regions involved in immune responses for culprit drugs such as clozapine, carbimazole and sulfasalazine [15,16,17,18,19].

Although the biological significance of many of the reported associations has only partially been unmasked, these findings would tend to favor the involvement of the immune system in these adverse drug reactions. Conversely, for metamizole, recent evidence from mechanistic in vitro studies is not in agreement with an immune-system driven mechanistic hypothesis as it has been shown that the main metamizole metabolite MAA (*N*-methyl-4-aminoantipyrine) reacts with the hemoglobin break down product hemin to produce a reactive intermediate that is cytotoxic for promyelocytic HL60 cells, but not mature neutrophil granulocytes [20,21,22]. Given this conflicting evidence, exploring genetic associations with MIA could be helpful to shed more light on whether to support or reject either hypothesis.

To identify loci associated with metamizole-induced neutropenia/agranulocytosis, we performed single-marker tests of association by means of a GWAS using genotype data from the largest MIA/MIN cohort to date (86 cases, 464 controls). Genetic variants in 84 candidate genes were analyzed in a Swiss discovery cohort (45 cases, 191 controls), with replication of findings and a genome-wide meta-analysis in two independent cohorts genotyped by the European Drug-induced Agranulocytosis Consortium (EuDAC).

## 2. Materials and Methods

### 2.1. Ethical Statement

This study was approved by the local ethics committees “Ethikkommission Nordwest- und Zentralschweiz” and “Kanton Bern” in Switzerland (EKNZ BASEC 2015–00231, KEK-Nr 2015-00231). Research was conducted according to the latest update of the Declaration of Helsinki. Collection of biological materials and clinical information was undertaken with a written informed consent from all participants. Samples and data of German (EuDAC-DE) and Spanish (EuDAC-ES) patient cohorts had previously been collected through the European Drug-induced Agranulocytosis Consortium (EuDAC) with approval by the local ethics committees (22 December 2014, Málaga, Spain; RTF011, Barcelona, Spain; Charité–Universitätsmedizin Berlin, Germany).

### 2.2. Study Design and Participants

This retrospective observational case-control study included genotype data from a Swiss discovery cohort (MIA/MIN-CH) that consisted of a total of 53 cases, 39 tolerant controls and 161 unexposed controls recruited at the University Hospitals Basel and Bern. MIN and MIA were defined by an absolute neutrophil count (ANC) below 1.5 × 10^9^ cells/L, and 0.5 × 10^9^ cells/L respectively. Tolerant controls included in the study had received at least 500 mg metamizole per day for a minimum of twenty-eight consecutive days, a treatment duration encompassing the latency time observed for a majority of cases for the occurrence of MIA based on a previous report [6]. Clinical drug tolerability during metamizole therapy was assessed by the absence of symptoms including fever, sore throat, or mucositis [23]. EDTA blood samples for genetic analyses were collected and coded at the time of recruitment. Clinical data, including patient characteristics and concomitant drug therapy (such as antibiotics, analgesics and β-lactam antibiotics) were retrieved from medical charts. A more detailed description of participants recruited for this study has been previously published [23]. In addition, participants genotyped as part of a project of the EuDAC were assigned to two independent replication cohorts based on their recruitment site: Germany (EuDAC-DE) and Spain (EuDAC-ES). Additional description of these cohorts is available in previous reports [16]. The majority of subjects were of self-reported European ancestry, which was consistent with multidimensional scaling (MDS) analyses of population structure using genome-wide genotype data (see further details below).

### 2.3. Genotype Data and Quality Control

For the samples of the MIA/MIN-CH discovery cohort, genomic DNA from whole blood or buffy coat was extracted and purified using the QIAamp DNA Blood Maxi or Midi Kits (Qiagen, Hilden, Germany) according to the manufacturer’s instructions. Eluted DNA was quantified with the Qubit 4 Fluorometer (ThermoFisher Scientific, Waltham, MA, USA) and quality was assessed by measuring absorbance at A230, A260 and A280 on a NanoDrop 1000 Spectrophotometer (ThermoFisher Scientific).

Genotyping in each of the contributing cohorts was conducted independently at different time points using various high-density SNP arrays (Figure 1). For the MIA/MIN-CH cohort, genotyping was performed using the Infinium Human CoreExome-24 BeadChip and processed by the iScan System together with a customized Tecan liquid-handling robot (both Illumina, San Diego, CA, USA) at the iGE3 Facility (University of Geneva, Geneva, Switzerland). Genotype calls were generated with the GenomeStudio software (Illumina). For replication and meta-analyses, existing genotype data was obtained through the EuDAC.

In all datasets, SNP positions were based on National Center for Biotechnology Information (NCBI, Bethesda, MD, USA) build 37 (hg19) and alleles were labeled on the positive strand of the reference genome. Similar genotyping quality control (QC) procedures [24,25,26,27,28] were used for each cohort using PLINK v1.9 [29]. Specifically, individual samples were removed if more than 2% of SNPs failed genotyping (call rate < 98%), and individual SNPs were removed if they showed a Hardy-Weinberg Equilibrium (HWE) *p*-value < 0.001 [30,31,32,33,34], a minor allele frequency (MAF) < 1%, if more than 2% of samples were flagged as having failed genotype calling for any given SNP (call rate < 98%) and if any duplicate or triplicate discordance was detected.

The rs111876221 polymorphism in the SERINC5 gene was additionally genotyped in the MIA-CH cohort using Polymerase Chain Reaction (PCR) TaqMan qPCR Master Mix 2× and a custom TaqMan SNP Genotyping assay kit (cat. 4331349, Assay ID-ANZTJD4, ThermoFisher Scientific), according to the manufacturer’s recommendations. Real-time PCR reactions were performed using 10 μL final reaction volumes, consisting of 5 μL 2× master mix, 4.75 μL water, 0.25 μL 40× probe assay and 1.0 μL gDNA. The amplification was performed with the QuantStudio™ 6 Flex System (ThermoFisher Scientific). The PCR program used was as follows: 10 min at 95 °C with at least 40× (15 s at 95 °C and 1 min at 60 °C). TaqMan assay results were validated (seven samples homozygous for the major allele, all samples with heterozygous genotype and other samples with an undetermined genotype) using Sanger sequencing. Details pertaining to the SNP primer sequences utilized for genotype validation can be found in Appendix A.

### 2.4. Multidimensional Scaling and Identification of Genetic Outliers

Multi-dimensional scaling (MDS) analysis was performed independently for each cohort using PLINK 1.9 [29]. SNPs that passed quality control were pruned using the linkage disequilibrium (LD) pruning parameters of r^2^ < 0.2 over a window size of 50. Genome-wide identity-by-descent (IBD) given identity-by-state (IBS) information was calculated using all SNPs that remained after pruning. Five nearest neighbors were identified for each individual based upon the pairwise IBS distance and the distance to the nth nearest neighbor was converted to a Z score. Fourteen individuals (nine in MIA/MIN-CH, two in EuDAC-ES and three in EuDAC-DE) with a minimum Z score less than −2 among the five nearest neighbors were excluded from analysis as population outliers. Furthermore, three pairs of EuDAC individuals (one in EuDAC-DE and two in EuDAC-ES) showed high levels of IBD sharing and one individual from each pair was randomly selected to be excluded from subsequent analyses due to this cryptic relatedness.

### 2.5. Imputation

Imputation of SNPs was performed by minimac3 [35,36,37] on the Michigan Imputation Server [38] using the Haplotype Reference Consortium (HRC 1.1 2016) reference panel [39] on the basis of pre-imputation SNP-level validated data (https://www.well.ox.ac.uk/~wrayner/tools/) from each cohort (Appendix A). HRC-based imputation was performed separately for each cohort, which increased the genome-wide SNP densities to 39′004′932, 38′994′811 and 39′008′189 SNPs for MIA/MIN-CH, EuDAC-ES and EuDAC-DE respectively. After performing quality control as described above, a total of 7′608′978, 7′675′043 and 7′422′709 SNPs were retained for the meta-analysis performed using a fixed-effects model in the MIA-CH, EuDAC-ES and EuDAC-DE cohorts respectively.

### 2.6. Candidate Gene and Genome-Wide Association Analyses

All analyses were carried out with PLINK (version 2.0) using logistic regression with the first four MDS dimensions and sex (only for EuDAC-ES) chosen as covariates. An additive genetic model was assumed for all variants and the genomic inflation factor was calculated for each analysis to assess potential confounding effects of population stratification [40,41].

Association analyses were carried out in multiple steps: Initial discovery analyses were carried out both in the entire MIA/MIN-CH cohort (*N* = 45 MIA/MIN cases, *N* = 191 tolerant/unexposed controls) and including only agranulocytosis cases of this cohort (*N* = 30 MIA cases, *N* = 191 tolerant/unexposed controls) focusing first on specific candidate gene sets followed by genome-wide association analyses. Candidate variants identified in the MIA/MIN-CH cohort were subsequently investigated for replication in the EuDAC cohorts. In a second discovery analysis, a genome-wide meta-analysis was carried out including association summary statistics from all three cohorts (Figure 1). In all analyses, tolerant and unexposed controls were combined as a control group in the MIA/MIN-CH cohort.

To perform candidate gene analyses, we analyzed different genes following a two-stage approach. First, eight genes were previously identified by association studies [15,42,43,44,45,46,47] of agranulocytosis induced by other drugs (*NOX3*, *SERINC5*, *NQO2*, *GSTM1*, *SLCO1B3*, *SLCO1B7*, *TAP2*, *AGER*; Phase I set) were selected and then 50 additional candidate genes (Phase II set) were included for their hypothesized relationship to drug-induced agranulocytosis based on their functional annotation potentially related to the oxidative metabolism of metamizole [48], the anti-oxidant defense [20,22] and the immune system defense response specific to genes expressed in granulocytes [49]. For each gene, we examined SNPs in a window of 10kb upstream and downstream of the gene. We used a statistical significance threshold *p*-value based on the method of Li et al. [34], which calculates the effective number of independent tests (M^e^) and applies a Bonferroni correction based on that number. A list of all genes part of the Phase II gene set is reported in a separate Appendix A.

For all genome-wide analyses, a threshold *p*-value of 1 × 10^−7^ was used to declare whether an association signal was statistically significant at a genome-wide level. This threshold was used based on an evaluation of empirical replication success of discovery GWAS, demonstrating good replication success for borderline associations with *p* < 1 × 10^−7^ [50]. The SNP association *p*-values from the three cohorts were subjected to meta-analysis with METAL using an inverse variance weighted scheme [51]. Identified candidate loci were further investigated for other traits associated with variation in these regions using publicly available databases of GWAS results (https://genetics.opentargets.org) and for effects on regulatory motifs or on expression from eQTL studies [52].

## 3. Results

### 3.1. Cohort Characteristics

Of the 96 cases of metamizole-induced agranulocytosis identified, ten were excluded after adjudication: eight cases in MIA/MIN-CH (two because they were identified as suggested genetic outlier i.e., nearest neighbor analyses suggesting non-European ancestry, two because of ongoing infectious diseases, four because of immunosuppressive therapy with cytotoxic drugs) and two cases in EuDAC-ES (because of suggested genetic outlier). No cases were excluded in the EuDAC-DE cohort.

Of the 479 controls identified, data from fifteen samples were excluded after adjudication: nine controls in MIA/MIN-CH (one individual was recruited twice as determined by IBD analysis, seven individuals because they were identified as suggested genetic outlier, and one because of sample genotype call rate < 50%), two controls in EuDAC-ES (because of cryptic relatedness) and four controls in EuDAC-DE (three because of suggested genetic outlier and one because of cryptic relatedness). Demographic characteristics of the tolerant and unexposed controls in the discovery cohort (MIA/MIN-CH) are presented in Appendix A. Comparison of demographic and clinical factors between the three independent cohorts are shown in Table 1. Cases were balanced by sex and age. Data on sex and age was unavailable for controls in the EuDAC-DE and EuDAC-ES cohorts.

### 3.2. Association Analyses in the Discovery Cohort (MIA/MIN-CH and MIA-CH)

#### 3.2.1. Candidate Gene Analyses

In the Swiss discovery cohort of 236 individuals, the genomic inflation factor (λ) was λ = 1.00 and no systemic bias was detected in any of the analyses that were conducted. The standard per-SNP significance threshold for a family-wise error rate (FWER) of α = 0.05 was estimated at 5.4 × 10^−4^ for the phase I candidate gene analysis and 1.08 × 10^−4^ for phase II. No statistically significant SNPs associated with MIA/MIN and MIA, respectively were identified in either phase I or phase II when including all 45 MIA and MIN cases (Appendix A) or only the 30 MIA cases (Appendix A). The top finding in both phase I analyses was a common intronic variant in the serine incorporator 5 (*SERINC5)* gene (rs10041917, *p* = 3.4 × 10^−3^, OR = 0.44, 95%-CI 0.25–0.76). This signal remained the top finding also in the analysis of the extended number of candidate genes in phase II. The rs10041917 variant in *SERINC5* was in linkage disequilibrium with a variant (rs111876221) previously reported to be associated with sulfasalazine-induced agranulocytosis (D’ = 1.0, *p* = 0.0068) as determined using the LDpair tool in LDlink using 1000 Genomes project data from European populations (https://ldlink.nci.nih.gov/?tab=ldpair). To evaluate a potential association with MIN/MIA, rs111876221 was separately genotyped in the entire MIA/MIN-CH cohort as this SNP was not included on the Infinium array and imputation results were only obtained in a later stage of the data analysis. Direct genotyping of rs111876221 in the MIA/MIN-CH cohort revealed that the A allele at that locus, which is linked to the A allele of rs10041917, was found more frequently in the case group (MAF = 0.055) compared to the controls (MAF = 0.016) indicating the same direction of effect. This association was shown to be significant with a one-sided Fisher’s exact test *p*-value of 0.03 (OR = 3.85, 95% 1.12–13.25).

Among the top SNPs from the phase I analysis in agranulocytosis cases only (Appendix A), several additional variants were identified as of potential interest. Specifically, the same missense SNP in the ATP binding cassette subfamily B member (*TAP2*) gene, rs2228391 (*p* = 5.1 × 10^−3^, OR = 377.7, 95% CI 5.93–24028), as observed here at a higher frequency in MIA cases was previously associated with susceptibility of a northern Chinese Han population to antithyroid drug-induced agranulocytosis [53]. In addition, another missense rs4148876 SNP in *TAP2* (*p* = 6.1 × 10^−3^, OR = 3.06, 95% CI 1.37–6.83) has been reported to be correlated with gene expression of the *HLA-DOB* and *TAP2* genes in several tissues including whole blood [54,55,56,57], Similarly, the intergenic variant rs6588432 SNP near the glutathione peroxidase 7 (*GPX7*) gene, has been reported to be associated with gene expression of *GPX7* and origin recognition complex subunit 1 (*ORC1L*) in several tissues including blood [54,57]. Due to their potential functional relevance, these additional variants were selected to assess replication in the EuDAC cohorts.

#### 3.2.2. Genome-Wide Association Analyses

We performed two genome-wide association analyses of 304,704 genotyped variants, first in the entire MIA/MIN-CH cohort (*N* = 45 cases, *N* = 191 tolerant/unexposed controls) and then including only MIA cases (*N* = 30 cases) versus the same 191 controls. None of the SNPs showed an association at the genome-wide significance level (*p* < 1 × 10^−7^) in these analyses (Figure 2). In the analysis including both MIA and MIN cases (Figure 2a), the leading SNP with the best evidence for a suggestive association was in an intergenic region on chromosome 6 (rs191786, *p* = 8.6 × 10^−6^, OR = 0.24, 95% CI 0.13–0.45) near the *TENT5A* gene. In the analysis with only MIA cases (Figure 2b), the leading SNP was also in an intergenic region on chromosome 6 (rs9366076, *p* = 2.89 × 10^−6^, OR = 4.81, 95% CI 2.49–9.30). 

Interestingly, this lead SNP was previously reported to be associated with gene expression of the nearby ribonuclease T2 (*RNASET2*) gene in multiple tissues [55,57,58] and in neutrophils [58]. Appendix A summarizes the leading associations of the genome-wide analyses.

#### 3.2.3. Replication in Independent Cohorts

To evaluate potential replication of these candidate associations in independent cohorts, directly typed and imputed genotype data from the EuDAC-ES and EuDAC-DE cohorts was used (Appendix A). Based on genome-wide association analyses for 1,786,726 genotyped SNPs in the EuDAC-ES cohort and 650,136 SNPs in the EuDAC-DE cohort, no evidence of systemic bias was detected in these datasets with genomic inflation factors (λ) of λ = 1.016 and λ = 1.047 in the EuDAC-ES and EuDAC-DE cohorts, respectively.

For the candidate SNPs selected from the phase I and phase II candidate gene analyses in the MIA/MIN-CH and MIA-CH cohort, we did not observe any significant associations with MIA in the two independent EuDAC cohorts (Appendix A). Due to reduced imputation accuracy in the EuDAC-ES (RSQ = 0.74) and EuDAC-DE (RSQ = 0.26) cohorts, replication of the rs111876221 variant previously associated with sulfasalazine-induced agranulocytosis could not be reliably assessed in those cohorts. Specifically, the reduced genotype imputation accuracy for this relatively uncommon SNP was illustrated in the MIA/MIN-CH cohort where the genotype was not correctly imputed for two out of eleven heterozygous minor allele carriers when compared to direct genotyping, in spite of an imputation RSQ of 0.91.

Associations of the leading SNPs from both genome-wide analyses in the MIA/MIN-CH and MIA-CH discovery cohort (Appendix A), rs191786 and rs9366076 near *RNASET2*, previously reported as an eQTL of this gene, could not be replicated in the two independent European cohorts (Appendix A).

### 3.3. GWAS Meta-Analysis across Independent Cohorts

For the meta-analyses, genome-wide association results from all three cohorts based on imputed genotype data were used. Also, in the meta-analysis, two analyses were conducted, first including MIN cases in the MIA/MIN-CH cohort and a second analysis only considering MIA cases. Manhattan plots of the results of both meta-analyses are shown in Figure 3. The first meta-analysis performed in all 86 cases and 464 controls across all three cohorts revealed that a leading SNP with the best evidence for association in an intronic region of the transforming growth factor β receptor 3 (*TGFBR3*) gene on chromosome 1 (rs11583606, *p* = 1.72 × 10^−7^, OR = 7.95% CI 3.37–14.5) (Appendix A). The second strongest independent signal was positioned in an intronic region in the protein tyrosine phosphatase receptor type O (*PTPRO*) gene on chromosome 12 (rs112917452, *p* = 9.92 × 10^−7^, OR = 7.3, 95% CI 3.29–16.20) (Appendix A). No association reached statistical significance at the genome-wide level (Table 2).

The second meta-analysis (Table 3) performed in the 71 MIA cases and 464 controls revealed a leading SNP with the best evidence for association in an intergenic region on chromosome 9 (rs55898176, *p* = 1.01 × 10^−7^, OR = 4.01, 95% CI 2.41–6.68) (Appendix A), only marginally not meeting the genome-wide significance threshold. This SNP was identified to be located near a long non-coding RNA with *CAAP1* as the closest coding gene at a distance of 177 kilobases to the canonical Transcript Start Site (TSS) of the gene (Figure 4).

The second strongest independent signal was positioned in an intronic region in the Sushi von Willebrand Factor Type A, EGF and Pentraxin Domain Containing 1 (*SVEP1*) gene, also on chromosome 9 (rs4427239, *p* = 5.75 × 10^−7^, OR = 5.47, 95% CI 2.81–10.65) (Appendix A). Several other genes located in proximity of *SVEP1* include the thioredoxin (*TXN*) gene, involved in the cellular antioxidant defense (Figure 4). An evaluation of the frequencies of the associated alleles for the lead signals from both meta-analyses in the MIN vs. MIA cases of the MIA/MIN-CH cohort is shown in Appendix A, respectively. Interestingly, the observed allele frequencies of the leading SNPs show that the risk allele distributions of rs11583606 and rs112917452 were higher in the MIN cases than in the MIA cases (Appendix A). In contrast, the risk allele distribution of rs55898176 and rs4427239 were higher in the MIA cases than in the MIN cases (Appendix A). Post stage GWAS analysis of these candidate loci using HaploReg [52] and GTEx [59] did not indicate any potential regulatory effects associated with the top SNPs across the available gene expression quantitative trait loci (eQTL) datasets. Furthermore, no GWAS lead variants were found to be linked to any of the candidate loci. However, several hematological measurement traits such as hemoglobin concentration, autoimmune hemolytic anemias and monocyte count reported by previous studies were found to be associated (*p* < 0.05) with the rs112917452, rs55898176 and rs4427239 variants in the UK Biobank and GWAS catalog summary statistics repository respectively. Also, several association studies [60] reported associations between lead variants in the *SVEP1* gene and hematological measurements obtained from blood plasma including neutrophil percentage of granulocytes (*p* = 4.5 × 10^−14^), white blood cell count (*p* = 2.0 × 10^−15^), platelet count (*p* = 2.2 × 10^−11^) and eosinophil percentage of granulocytes (*p* = 9.1 × 10^−17^).

## 4. Discussion

We performed candidate gene-based and genome-wide association analyses for metamizole-induced agranulocytosis in three datasets totaling 86 MIN/MIA cases and 464 controls. To our knowledge, this is the first study to investigate genetic associations with MIA at a genome-wide level in the largest patient cohort available to date. In contrast to previous GWAS for agranulocytosis related to other pharmaceutical agents, we did not identify any genome-wide significant genetic associations that would implicate an immune-mediated mechanism for MIA. However, we report suggestive evidence for an association of two loci on chromosome 9, one of which implicates anti-oxidant components as well as hematopoiesis, that could serve as plausible candidates for studies to come.

Overall, the genetic susceptibility to rare yet potentially lethal adverse drug reactions, ranging from skin and liver injury to agranulocytosis, has been repeatedly shown to be specific to each offending drug [61,62,63,64,65]. Nevertheless, common underlying T cell-mediated mechanisms have been identified for some drug-induced serious skin reactions and drug-induced liver injury. Similarly, the strongest pharmacogenetic associations pertaining to drug-induced agranulocytosis up to date mostly involved immune-related genes such as specific HLA alleles [15,16,17,44,45,46,66,67], even though these findings have yet to be corroborated by mechanistic studies. Here, on the other hand, in line with a previous association study focused on classical HLA genes in the same cohort [68], none of the candidate loci identified as leading markers in our genome-wide meta-analysis involved genes that would support an immune-mediated mechanism underlying MIA. These findings thus suggest that the underlying mechanism for MIA may differ from other agranulocytosis-inducing drugs. Such a lack of evidence for an immune-mediated mechanism underlying MIA is concordant with recent in vitro studies suggesting that the cytotoxicity in immature myeloid cells stems from a reactive electrophilic intermediate formed from the main metabolite of metamizole, MAA and hemin [20,21,22]. Genetic variation in enzymes involved in the heme degradation and antioxidant defense pathways could thus impact an individual’s susceptibility to develop MAA-associated myelotoxicity.

In this context, the suggestive association of the SNP rs4427239 (chr9p13) in the analysis of only MIA cases vs. tolerant/population controls is notable due to the reported associations of variants at this locus with hematopoietic traits and the role of its nearby gene *TXN*, encoding for thioredoxin (TXN), in redox control [69,70,71,72,73,74,75]. As part of the thioredoxin system, TXN is an antioxidant enzyme found in a variety of cells including granulocytes [76,77], and is involved in the defense against oxidative stress by controlling cellular free radicals and reactive oxygen species [78,79,80,81]. Functional investigations of TXN in vitro cell culture models, in particular involving granulocyte precursor cells may thus be of interest [20,22]. While mRNA expression of *SVEP1,* the gene in which the lead SNP rs4427239 is located, was not detected in any immune cells, it was reported to be overexpressed in a hematopoiesis-supporting splenic stromal cell line [82]. Furthermore, its encoding protein SVEP1 was identified as a ligand of integrin α 9 β 1 (ITGA9) [83], which is involved in signal transduction in hematopoietic stem cells [84], and for which an important role has been suggested for granulopoiesis [85]. Given these findings, *SVEP1* also represents an interesting starting point for experimental investigations aiming to clarify the impact of metamizole and its metabolites on cell adhesion between granulocyte precursors and stromal cells during maturation.

While the signal related to rs4427239 thus includes two nearby genes with biological plausibility, the strongest association signal we observed in the meta-analysis including only MIA cases was the intergenic variant rs55898176 (chr9p13) located near an uncharacterized long non-coding RNA (LOC105376000). The nearest coding gene to this variant is *CAAP1*, which encodes conserved anti-apoptotic proteins that modulate the mitochondrial apoptosis pathway [86]. The possible biological implications of this association with respect to MIA thus still remains to be elucidated.

Interestingly, the frequencies of both of the above-mentioned lead SNPs, rs4427239 and rs55898176, were substantially lower in the Swiss MIN cases compared to MIA cases and similar to the frequencies observed in controls (Appendix A). The allele frequencies of these SNPs in the MIN cases thus did not follow an intermediate distribution between the frequencies in the MIA cases and the controls, as has been observed for genetic associations with other adverse drug reactions for cases with intermediate severity [87]. If these signals indeed represent true causal associations, a potential underlying mechanism involving the intergenic locus on chromosome 9, or the *SVEP1* locus in agranulocytosis may not apply to metamizole-induced neutropenia.

Conversely, we observed that the frequencies of the risk alleles for the rs11583606 and rs112917452 SNPs identified as leading markers the meta-analysis of all cases versus controls were substantially higher in the Swiss MIN cases compared to the Swiss MIA cases (Appendix A). These signals thus appear to be driven primarily by the MIN cases in the Swiss cohort. The leading loci from this meta-analysis, *TGFBR3* and *PTPRO* thus showed less consistent effects among the MIA cases across the three cohorts and as a consequence may not represent the most promising signals for further investigation. Similarly, both candidate gene analyses (phase I and phase II) revealed leading markers in the Swiss cohort positioned in intronic regions of the same gene, *SERINC5*, a gene coding for transmembrane proteins reducing Human Immunodeficiency Virus (HIV-1) infectivity [88]. In this gene, a genome-wide significant intronic SNP rs111876221 was previously reported to be associated with sulfasalazine-induced agranulocytosis [15]. However, given that the associations in this gene were not replicated in the Spanish or German cohorts and the lack of obvious functional implications of the *SERINC5* gene as it relates to MIA, we do not consider this signal as a strong candidate for further investigation.

None of the identified lead SNPs were located in protein-coding regions or were found to be eQTLs in previous studies including the 48 human tissues analyzed in the eQTL analysis of the Genotype-Tissue Expression (GTEx) project [59]. Importantly, however, it should be considered that the biological interpretation of the candidate loci potentially associated with MIA is hampered by the scarcity of functional data on the impact of genetic variation on gene expression specifically in bone marrow or in granulocyte precursors, where cytotoxic effects of metamizole may take place [20,22]. As a consequence, potential regulatory effects of these candidate SNPs at the level of the bone marrow and specifically during granulocyte maturation cannot be excluded. With the increasing availability of single-cell RNA sequencing studies providing data at the level of individual cell types or cell differentiation stages [89,90,91], additional functional annotation of the identified candidate loci may be an option in the future.

Main strengths of this study are the fairly homogenous discovery cohort and the possibility of replication in a second and third cohort. Although the case-control cohorts originated from three different European countries, by limiting our analyses to individuals of European ancestry using genetic information and adjusting for residual population structure, the likelihood of false positive associations due to population stratification was reduced. Also, by adhering to the case definition criteria of the EuDAC studies for the selection of individuals of the MIA-CH cohort, the three cohorts were similar in their definition of the ADR phenotype, which provided the advantage of increased power for the genome-wide meta-analysis.

Although this study has several strengths, it should also be interpreted in the context of several limitations. First, with the limited sample size, our study was not powered to detect associations with smaller effect size for uncommon variants. Despite the assembly of large patient cohorts by large international consortia, studies on rare idiosyncratic adverse drug reactions with incidences of 1 in 10,000 patients or less, finding it difficult to recruit large enough numbers of cases and tolerant comparison controls, have been faced with the reality of possibly being underpowered and to date have commonly included less than hundred cases [92]. Nevertheless, several of these studies have successfully identified associations due to large effect sizes being commonly observed for adverse drug reactions compared to genetic associations with common diseases [87,92,93,94,95]. To try to attenuate this limitation, recruitment of more cases and tolerant controls would have been a wishful but very challenging endeavor given the rarity of MIA. Second, another limitation lies in the use of population controls instead of drug-tolerant controls. However, the use of population controls instead of tolerant controls in genetic association studies is suitable for rare ADRs (incidence < 1%) as is the case for MIA [96]. Third, any residual selection bias stemming from the different recruitment channels for cases and controls cannot be fully ruled out, even though no systematic genetic differences between cases and controls were identified by MDS. Finally, considering the lack of statistically significant associations and the lack of an additional independent replication cohort to further investigate the lead signals from the genome-wide meta-analysis, it is possible that some of the identified candidate loci, or possibly even all, represent false positive associations, despite their consistent effects observed in the three investigated cohorts. Similarly, genetic variation associated with MIA should be explored in other populations as well, as the frequency of the identified variants and their clinical impact could vary between different ethnicities.

## 5. Conclusions

In this study, we associated genotyped and imputed markers with MIA in three independent cohorts of European individuals. Taken together, although our findings do not have immediate clinical utility, they point towards novel candidate genes with a possible role in the mechanism of MIA in individuals of European ancestry. Interestingly, our findings do not implicate biological pathways that have been previously identified in association studies for other causal drugs of agranulocytosis studied in populations of the same ethnicity, for which current evidence is consistent with an immune-mediated mechanism. Although further studies are warranted to follow up on these suggestive candidate genes and pathways, our findings are a stepping-stone in the ongoing endeavor of better understanding the genetic background behind MIA and its underlying causal mechanisms. Ultimately, this could pave the way to a better screening or treatment for a safer use of metamizole.

## Figures and Tables

**Figure 1 genes-11-01275-f001:**
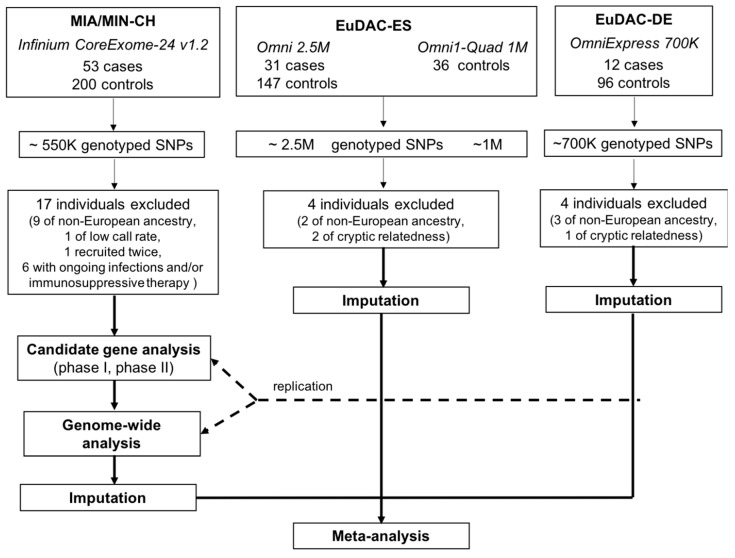
**Study design**. Cases and controls were genotyped separately using the indictaed Illumina arrays. MIA/MIN-CH = Swiss cohort, EuDAC-ES = Spanish cohort, EuDAC-DE = German cohort, SNP = single nucleotide polymorphisms, K = 1000, M = 100,000.

**Figure 2 genes-11-01275-f002:**
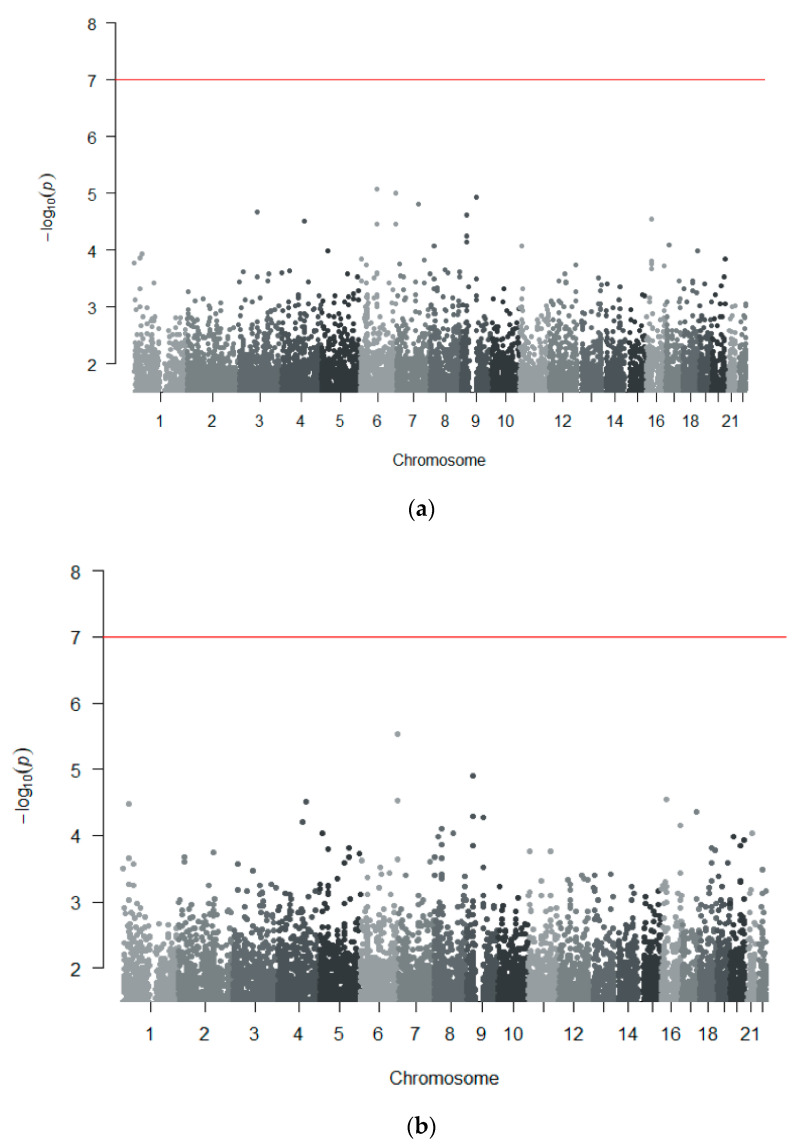
**Manhattan plot of genome-wide association analyses in the MIA/MIN-CH cohort**. Results from 304,704 genotyped SNPs after quality control are shown, adjusted by multidimensional scaling (MDS) dimensions 1–4: (**a**) Analysis of all 45 cases (MIA and MIN) versus 191 controls (tolerant and unexposed combined). The top SNP was rs191786, located in an intergenic region nearest to the *TENT5A* gene on chromosome 6; (**b**) Analysis of 30 MIA cases versus 191 tolerant/unexposed controls. The top SNP was rs9366076, located in an intergenic region nearest to the *RNASET2* gene on chromosome 6. The red line shows the threshold for genome-wide significance of 1 × 10^−7^.

**Figure 3 genes-11-01275-f003:**
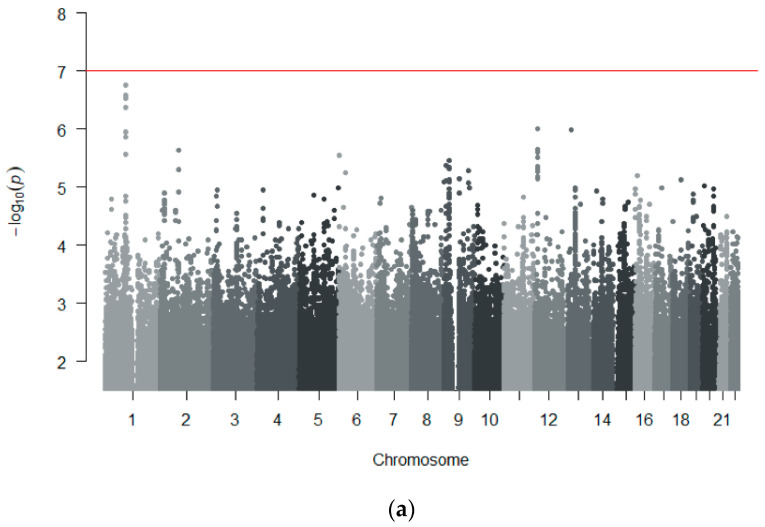
**Manhattan plot of GWAS meta-analyses in the three independent cohorts** (**MIA/MIN-CH, EuDAC-ES and EuDAC-DE**)**.** Results from approximately 7 million SNPs after imputation and quality control are shown, adjusted by multidimensional scaling (MDS) dimensions 1–4 and sex (only for EuDAC-ES). The red line shows the threshold for genome-wide significance of 1 × 10^−7^. (**a**) Analysis of all 86 cases (MIA and MIN) versus 464 controls (tolerant and unexposed). The top SNP was rs11583606, located in an intron of the *TGFBR3* gene on chromosome 1. (**b**) Analysis of 71 agranulocytosis cases (MIA) versus 464 controls (tolerant and unexposed). The top SNP was rs55898176, located in an intergenic region nearest to the *CAAP1* gene on chromosome 9.

**Figure 4 genes-11-01275-f004:**
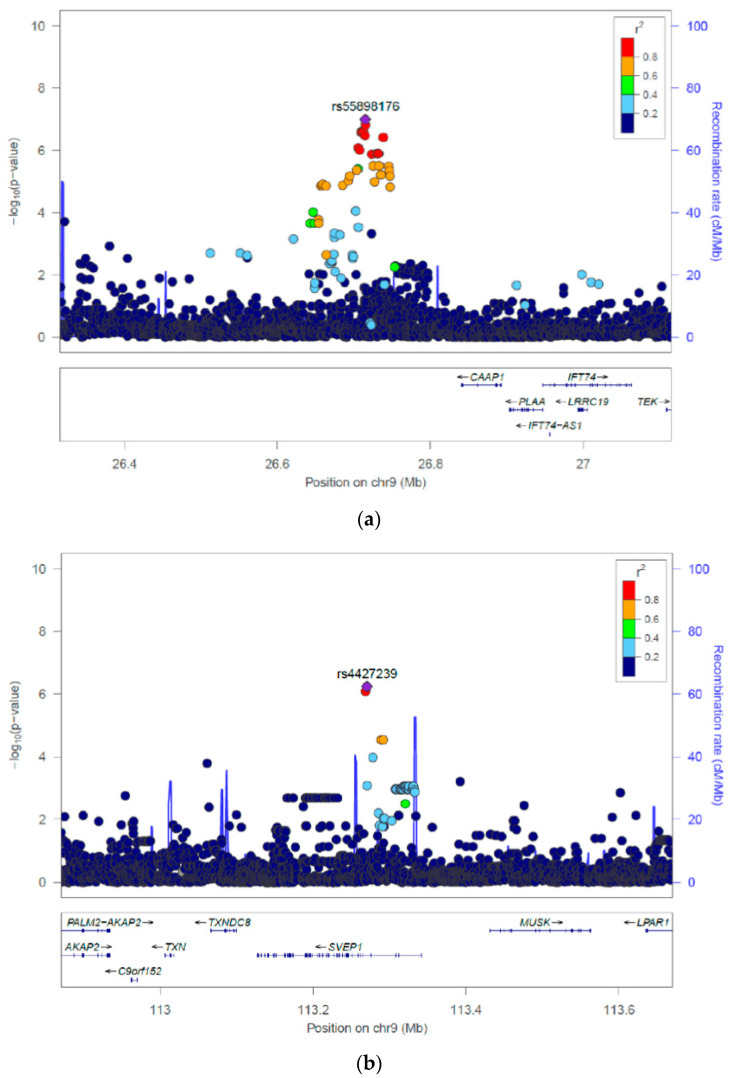
**Regional association plots for regions around rs55898176 and rs4427239**. Plots were produced in Locus Zoom and show the most strongly associated independent SNPs: rs55898176 (**a**) and rs4427239 (**b**) from the GWAS meta-analyses of the 71 MIA cases versus 464 controls (tolerant and unexposed). Different colors represent the strength of the LD of each SNP with the most significant SNP represented by a diamond.

**Table 1 genes-11-01275-t001:** **Characteristics of cases and controls in the three independent cohorts.** Data are N(%) or median (range). ^a^ cases, ^b^ controls, * latency missing for 3 MIA cases and 1 MIN case. ANC = lowest absolute neutrophil count, BMI = body mass index, NA = not applicable/not available.

Cohort	MIA/MIN-CH		EuDAC-DE		EuDAC-ES	
	Cases	Controls	Cases	Controls	Cases	Controls
	*N* = 45	*N* = 191	*N* = 12	*N* = 92	*N* = 29	*N* = 181
ANC < 500/uL	30 (67)	-	12	-	29	-
Sex, male (%)	13 (42)	17 (45)	4 (33)	41 (44.5)	6 (21)	87 (48)
Age, years (%)						
<25	7 (23)	1 (3)	2 (16.6)	NA	3 (10.3)	NA
25–44	11 (35)	6 (16)	6 (50)	NA	6 (21)	NA
45–64	10 (32)	15 (39)	2 (16.6)	NA	12 (41.4)	NA
65–74	3 (10)	9 (24)	2 (16.6)	NA	5 (17)	NA
>74	-	7 (18)	-	NA	3 (10.3)	NA
BMI, median (range)	24 (19–47)	28 (16–39)	NA	NA	NA	NA
Latency time * ^a^/treatment duration ^b^, days	17 (1–204)	25 (1–5297)	33.5 (4–9855)	NA	11.5 (1–235)	NA

**Table 2 genes-11-01275-t002:** **Top associations of the GWAS meta-analysis with MIA/MIN in all cases versus controls in the three independent European cohorts.** Top GWAS meta-analysis results based on approximately 7 million SNPs after imputation in 83 MIA/MIN cases vs. all 464 controls (tolerant and unexposed). All results were adjusted for four genetic multidimensional scaling (MDS) components and sex (only for EuDAC-ES). Chromosomal location is according to the Genome Reference Consortium human assembly GRCh37. CHR = chromosome, SNP = single nucleotide polymorphism, BP = base pair, MAF = minor allele frequency, OR [95%] = odds ratio with 95% confidence interval, HetISq = I^2^ statistic which measures heterogeneity on a scale of 0–100%.

CHR	SNP	Alleles (Minor/Major)	BP	MAF Cases MIA/MIN-CH| EuDAC-ES| EuDAC-DE	MAF Controls MIA/MIN-CH| EuDAC-ES| EuDAC-DE	OR [95%]	***p*** **-Value**	**Gene Region**	**HetISq**
1	rs11583606	T/C	92349247	0.10| 014| 0.083	0.023| 0.025| 0.027	7.0 [3.37–14.5]	1.72 × 10^−7^	*TGFBR3*	0
1	rs149072800	C/T	92445720	0.089| 0.12| 0.083	0.020| 0.019| 0.022	7.81 [3.57–17.1]	2.66 × 10^−7^	*BRDT*	0
1	rs146378328	G/A	92528047	0.089| 0.10| 0.083	0.018| 0.019| 0.021	8.31 [3.69–18.7]	2.93 × 10^−7^	*EPHX4*	0
1	rs75499485	G/A	92486274	0.089| 0.10| 0.083	0.020| 0.019| 0.021	7.96 [3.56–17.8]	4.23 × 10^−7^	*EPHX4*	0
12	rs112917452	C/A	15638858	0.067| 0.15| 0.042	0.016| 0.027| 0.016	7.30 [3.29–16.20]	9.92 × 10^−7^	*PTPRO*	0
12	rs118135416	A/G	15638914	0.067| 0.15| 0.042	0.016| 0.027| 0.016	7.30 [3.29–16.20]	9.92 × 10^−7^	*PTPRO*	0
12	rs7135120	T/C	15626920	0.067| 0.15| 0.042	0.016| 0.027| 0.016	7.30 [3.29–16.20]	9.92 × 10^−7^	*PTPRO*	0
12	rs143843248	T/C	15633812	0.067| 0.15| 0.041	0.016| 0.027| 0.016	7.30 [3.29–16.20]	9.92 × 10^−7^	*PTPRO*	0
13	rs73163933	A/G	33968020	0.12| 0.086| 0.21	0.036| 0.027| 0.054	5.36 [2.73–10.5]	1.01 × 10^−6^	*STARD13*	0
1	rs78201766	G/A	92379078	0.10| 0.14| 0.083	0.026| 0.030| 0.027	5.65 [2.81–11.34]	1.12 × 10^−6^	*TGFBR3*	0

**Table 3 genes-11-01275-t003:** **Top associations of the GWAS meta-analysis in MIA cases versus controls in the three independent European cohorts.** Top GWAS meta-analysis results based on approximately 7 million SNPs after imputation in 71 MIA cases vs. all 464 controls (tolerant and unexposed). All results were adjusted for four genetic multidimensional scaling (MDS) components and sex (only for EuDAC-ES). Chromosomal location is according to the Genome Reference Consortium human assembly GRCh37. CHR = chromosome, SNP = single nucleotide polymorphism, BP = base pair, MAF = minor allele frequency, OR [95%] = odds ratio with 95% confidence interval, HetISq = I^2^ statistic which measures heterogeneity on scale of 0–100%.

CHR	SNP	Alleles (Minor/Major)	BP	MAF Cases MIA/MIN-CH| EuDAC-ES| EuDAC-DE	MAF Controls MIA/MIN-CH| EuDAC-ES| EuDAC-DE	OR [95%]	***p*-Value**	**Gene Region**	**HetISq**
9	rs55898176	T/C	26715294	0.27| 0.24| 0.25	0.065| 0.12| 0.081	4.01 [2.41–6.68]	1.01 × 10^−7^	-	21.7
9	rs112223975	C/G	26715828	0.27| 0.24| 0.25	0.065| 0.12| 0.081	3.89 [2.34–6.48]	1.50 × 10^−7^	-	28.6
9	rs11790418	G/A	26713012	0.27| 0.24| 0.25	0.065| 0.12| 0.098	3.81 [2.29–6.35]	2.54 × 10^−7^	-	29.3
9	rs1434481	G/C	26711134	0.27| 0.24| 0.25	0.065| 0.11| 0.098	3.81 [2.29–6.35]	2.54 × 10^−7^	-	29.3
9	rs28475568	G/C	26709933	0.27| 0.24| 0.25	0.065| 0.11| 0.098	3.81 [2.29–6.35]	2.54 × 10^−7^	-	29.3
9	rs28649995	A/G	26709912	0.27| 0.24| 0.25	0.065| 0.11| 0.098	3.81 [2.29–6.35]	2.54 × 10^−7^	-	14.6
9	rs56285046	A/G	26714950	0.28| 0.24| 0.25	0.073| 0.12| 0.092	3.70 [2.27–6.05]	3.25 × 10^−7^	-	0
9	rs77949268	A/G	26738366	0.27| 0.26| 0.25	0.078| 0.12| 0.10	3.59 [2.20–5.87]	3.74 × 10^−7^	-	0
9	rs4427239	A/G	113270601	0.16| 0.15| 0.041	0.029| 0.041| 0.016	5.47 [2.81–10.65]	5.75 × 10^−7^	*SVEP1*	0
9	rs10759436	C/T	113268650	0.15| 0.15| 0.041	0.029| 0.041| 0.016	5.81 [2.92–11.54]	8.13 × 10^−7^	*SVEP1*	0

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
