# Peer review of "Genome-Wide Association Study of Metamizole-Induced Agranulocytosis in European Populations"

_genes, 2020, doi:10.3390/genes11111275_

Round 1

Reviewer 1 Report

The authors have addressed my questions and there are only few minor changes

Minor comments:

  1. Simplify the second sentence of the introduction, from line 68 to 73
  2. The acronims for the terms MIN and MIA are define several times. They should be explained the first time they appear in the manuscipt
  3. According to the analysis well described along the manuscrip adjusting the results by sex, the authors should change the word gender by sex in table 1, where they described the characteristics of the population. Just to clarify, gender refers to the socially constructed roles, behaviours, expressions and identities of people. On the other hand, sex refers to a set of biological attributes associated with physical and physiological features including chromosomes, and gene expression.

Reviewer 2 Report

Authors present a multi-centre GWAS + meta-analysis of metamizole-induced agranulocytosis and neutropenia with associations of suggestive evidence, pointing out at some hematopoiesis checkpoints. In general, the study is of high relevance and appropriate design. However, I recommend that some extensive proofreading and clarification of several issues should be done to make the manuscript publishable.

Abstract: add sample sizes

Page 2, lines 83-86 – The last sentence defiitely needs revision, because it is unclear what Authors have intended to say.

Statistics. Authors should explain why p-value of 1x10-7 has been selected as the genome-wide significance threshold, not 5x10-8 or lower.

In all sections (Methods, Results, Discussion) it would be useful to see more clear description of the results, i.e. whether all the analysi included MIA + MIN combined group as cases that were compared with either exposed, or all controls (if I understood correctly).

Supplementary Figures and Tables need proofreading and formatting (very hard to look at them now)

Author Response

This manuscript is a resubmission of an earlier submission. The following is a list of the peer review reports and author responses from that submission.

Round 1

Reviewer 1 Report

I have reviewed the article by Cismaru and colleagues “Genome-wide association study of metamizole3 induced agranulocytosis in European populations”. In this article, the authors studied the association between genetic variants to metamizole-induced agranulocytosis (MIA) and neutropenia (MIN), following several approaches including candidate-gene analysis, genome-wide and meta-analysis. Despite the numerous analysis performed, the authors did not find any variant which reached genome-wide p-values. However, they found some less significant association worth to mention.

Major comments

Introduction

  1. In the abstract is mention that also neutropenia (MIN) is analyzed, but in the introduction the authors do not mention this variable. At least the difference between MIA and MIN and/or the definition of such variables should be mention.

Methods

  1. Study design and participants (line 109): they should indicate what the clinical variable is for aganulocytosis analysis. In Table 1 they mention they are analyzing ANC without more explanation.
  2. Explain differences between patients with MIA and MIN
  3. Genotype data and quality control, lines 139-140: Why a p-value <0.001 is selected for HWE? it is related to the number of SNPs analyzed?
  4. The number of patients and controls in figure 1 does not correspond with the numbers explained in the results section “cohort characteristics”. For example 10 patients and 15 controls are removed (a total of 25), but in the figure there are a total of 19.
  5. In pharmacogenetic studies like this one, a case-only strategy is followed to find significant associations. Comparing patients who take the drug and develop ADRs against those who take the drug and do not develop ADRs could find other variants with a higher association with agranulocytosis. The authors should justify the selection of a case-control strategy instead of case-only.

Results

  1. Along the results section several references are provide while describing the results (for instance line 241 and 253. The results should be a description of the authors´ results, without mentioning other works (since this is the aim of the discussion).
  2. (3.2.1) Candidate-gene analysis (lines 248 and 250): when the significant SNPs are mentioned, the nearest gene or the exact location should be also mentioned in the text like they do for rs6588432 in line 253.
  3. (3.2.2) Genome-wide association analysis (line 160): What is the difference between MIA and MIN patients? MIA=30 and MIN=45? In the methods section this should be explained. Also, in 3.2.1 candidate gene analysis, two types of analysis are specified with “all cases” (line234) and “only agranulocytosis cases”, do they correspond to MIA and MIN cases? This part should be better explained since it is also mention in lines 343 and 344.
  4. (3.2.2) Genome-wide association analysis (line 264): the nearest gene should be mentioned also in the text, not only in figure 2 legend (TENT5A).
  5. I would recommend to use the database GTEX (https://www.gtexportal.org/home/) to study if the SNPs mentioned along the results section are involved in changes in expression (for those with no eQTL data)
  6. There are errors in the format of the supplementary tables

Discussion

  1. In the last paragraph the strengths and limitations should be well differentiate, clarifying first the strengths and then the limitations.
  2. The design of the study (case-control instead of case-only) should also be mentioned as a limitation.

Minor comments

  1. References along the manuscript are in different formats (line 241 vs 253)
  2. The name of the gene TGFBR3 in line 302 and legend in figure 3 (line 314) are different.
  3. A revision of the English grammar should be done
  4. I would recommend to simplify some sentences because sometimes the message is lost (for instance, the sentence from line 445 to line 450 is extremely long and no clear at all)

Reviewer 2 Report

This paper describes a GWAS and candidate gene analysis of metamizole-induced agranulocytosis.  This is a rare, adverse drug reaction with expected pharmacogenomic effects.  The paper is generally well written and the study is sound.

Unfortunately, the findings are modest, which is likely due to the very small sample size.  I think it is still important to report such negative results; however, some details in the manuscript could be improved to justify some of the decisions made and conclusions drawn.

  1. In the abstract, it might be good to mention the replication results for the two candidate loci reported.
  2. As indicated in the introduction, the sample sizes are very small.  This is often the case for rare ADRs.  I think it would be worth adding text and citations to this effect either in the intro or discussion or both.  With the lack of statistically significant findings, readers could be very quick to dismiss the paper.  However, especially with rare ADRs, it is good to get them published so that as others accumulate cohorts, they can be combined with existing published data.
  3. In section 2.2, the authors indicate that controls did not have an ADR within 28 days.  Is this the time frame that cases would expect to experience the ADR?
  4. The HWE p-value threshold selected is very conservative.  Most GWAS use p<10-6.
  5. Justification for the direct genotyping of rs111876221 is needed.  Was this not captured on the GWAS array? Or imputed?  What made the authors select this SNP?  This seems to be a random SNP to add to the study.  Justification or motivation for this is needed.
  6. Why was the p-value threshold of 1x10-7 selected for the GWAS?  The standard in the field is 5x10-8.  While nothing is significant at either threshold, I still think you should use the industry standard.  It seems odd to select a different threshold without justification.
  7. The authors indicate that sex is not available in the EuDAC-DE or EuDAC-ES cohorts.  Couldn't this be inferred from the GWAS data?  
  8. In line 288, the authors indicate that the imputation accuracy was low.  I would not say that an Rsq of 0.75 is low.  That is actually pretty high.
  9. The authors spend a lot of time in the discussion talking about the suggestive findings.  I am not sure this is a good use of space given the lack of significance of the findings.